# Diagnostic Efficacy of Serological Antibody Detection Tests for Hepatitis Delta Virus: A Systematic Review and Meta-Analysis

**DOI:** 10.3390/v15122345

**Published:** 2023-11-29

**Authors:** Zhenzhen Pan, Sisi Chen, Ling Xu, Yao Gao, Yaling Cao, Zihao Fan, Yuan Tian, Xiangying Zhang, Zhongping Duan, Feng Ren

**Affiliations:** 1Beijing Institute of Hepatology/Beijing Youan Hospital, Capital Medical University, Beijing 100069, China; panzhenzhen0@163.com (Z.P.); 15764235913@163.com (L.X.); gaoyao@mail.ccmu.edu.cn (Y.G.); 18738337870@163.com (Y.C.); fzh_925@163.com (Z.F.); tianyuan2716@163.com (Y.T.); zhangxy2018@ccmu.edu.cn (X.Z.); 2Beijing Youan Hospital, Capital Medical University, Beijing 100069, China; thoughtfmu@163.com (S.C.); duan2517@ccmu.edu.cn.com (Z.D.)

**Keywords:** hepatitis delta virus, antibody detection, serological testing, diagnostic performance, meta-analysis

## Abstract

Background and Aims Coinfection of hepatitis delta virus (HDV) with hepatitis B virus (HBV) causes the most severe form of viral hepatitis, and the global prevalence of HDV infection is underestimated. Although serological testing of anti-HDV antibodies is widely used in the diagnosis of HDV, its diagnostic efficacy remains unclear. This study aimed to evaluate the diagnostic efficacy of HDV serological tests, the results of which may assist in the diagnosis of HDV. Methods Preferred Reporting Items for Systematic Reviews and Meta-analyses (PRISMA) guidelines were followed. The PubMed, Web of Science and Cochrane Library databases were searched from the beginning to 31 May 2023. Study quality was assessed using the Quality Assessment of Diagnostic Accuracy Studies-2 (QUADAS-2) tool. STATA SE was used for the meta-analysis of the sensitivity, specificity, positive likelihood ratio and negative likelihood ratio. Results Among a total of 1376 initially identified studies, only 12 articles met the final inclusion criteria. The pooled sensitivity and specificity were 1.00 (95% CI: 0.00–1.00) and 0.71 (95% CI: 0.50–0.78) for HDV total antibodies, 0.96 (95% CI: 0.83–0.99) and 0.98 (95% CI: 0.82–1.00) for anti-HDV IgM and 0.95 (95% CI: 0.86–0.98) and 0.96 (95% CI: 0.67–1.00) for anti-HDV IgG. The pooled sensitivity and specificity for HDV serological tests were 0.99 (95% CI: 0.96–1.00) and 0.90 (95% CI: 0.79–0.96). Conclusions This meta-analysis suggests that serological tests have high diagnostic performance in detecting antibodies against HDV, especially in HDV IgM and IgG. However, this conclusion is based on studies of a limited number and quality, and the development of new diagnostic tools with higher precision and reliability is still necessary.

## 1. Introduction

Hepatitis delta virus (HDV) is a blood-borne pathogen that relies on the envelope protein of HBV for the assembly and release of infectious virus particles [1,2]. HDV particles are composed of HBV envelope proteins surrounding the nucleocapsid, which is composed of a single-stranded circular RNA genome and viral HDV–antigen complex. HDV infection causes hepatitis D [3]. The clinical presentation of hepatitis D ranges from mild disease to fulminant liver failure [4]. There are two modes of clinical HDV infection: “coinfection” and “superinfection”. Coinfection refers to simultaneous HBV and HDV infection in individuals who have not previously been exposed to HBV and HDV. In adults, HDV/HBV co-infection is usually short-lived and self-limited. Studies have shown that HDV/HBV infection often leads to more serious consequences than HBV virus infection alone. Nevertheless, there are many individuals infected with HBV in the absence of HDV; when such an individual is exposed to HDV, it is referred to as superinfection. This pattern of infection causes severe acute hepatitis, which may be self-limiting but in most cases (up to 80%) progresses to chronic [5]. Once chronic HDV infection is identified, it usually aggravates pre-existing chronic hepatitis B. Patients who are infected with both HDV and HBV can usually eradicate both pathogens, while chronic HBV carriers who later become infected with HDV can develop chronic HDV infection and more severe liver damage [6]. Although HDV can inhibit HBV replication, HDV-related chronic hepatitis is frequently associated with severe necroinflammation and rapid progression to advanced stages of liver fibrosis and cirrhosis. Chronic HDV and HBV infection may also be associated with a higher risk of portal hypertension, hepatocellular carcinoma (HCC) and all-cause mortality than chronic HBV mono-infection [4,7].

According to a recent meta-analysis, an estimated 12 million people worldwide have been infected with HDV [8]. However, due to large gaps in diagnosis, especially in high-prevalence areas and populations, this number might be underestimated, which is supported by meta-analyses indicating that 50–72 million HBV carriers may be coinfected with HDV [9,10]. The exact prevalence and estimated number of HDV patients is still a subject of debate for several reasons, including unreliable assessment of infection and a lack of real-world screening [11]. In view of the significantly increased risk of adverse clinical outcomes (such as liver cirrhosis, HCC, etc.) in patients with HBV/HDV coinfection, increasing screening and early detection of HDV infection is the key to optimizing clinical treatment and reducing morbidity.

HDV infection refers to the replication of viral RNA with expression of the HD antigen (HD-Ag) and the specific immune responses of the host. HDV induces innate and adaptive immune responses in infected hosts, stimulating the production of immunoglobulin M (IgM) and immunoglobulin G (IgG) [1]. Hence, diagnostic tests for HDV fall into two main categories: (a) molecular tests for viral RNA and (b) serological tests for anti-HDV antibodies. Detection of viral RNA is widely used as the reference standard for the diagnosis of HDV because of high specificity and sensitivity. A previous study conducted by our research group evaluated the diagnostic performance of HDV RNA detection using the HDV RNA detection method and found that HDV RNA detection had high diagnostic efficacy [12]. However, the diagnostic efficacy of HDV serology has not been systematically evaluated and reported. RNA molecular detection has many limitations: (a) it requires trained technicians in a certified laboratory, (b) a long time is usually needed to generate results and (c) the highly sensitive equipment required for RNA molecular detection is expensive [13]. Serological testing mainly includes detection of IgG and IgM antibodies against HDV. Anti-HDV IgG positivity is found in patients in acute remission of HDV infection and chronic HDV infection, persisting for a long time after virus clearance. Conversely, the anti-HDV IgM antibody is detectable within 2–3 weeks of symptom onset and disappears 2 months after acute infection [14], though anti-HDV IgM is also elevated in patients with chronic HDV during disease flares. In addition, since there are eight genotypes of HDV, genes play an important role in the production and function of antibodies, and the coding sequence of genes determines the structural and functional properties of antibodies. Different combinations and arrangements of gene fragments produce different antibody types and subtypes. If the HDV antibody detection method is established for a common sequence of multiple HDV genotypes, then the HDV genotype will not affect the accuracy of the HDV antibody detection; if the HDV antibody detection method is established for a certain HDV genotype, it will affect the accuracy of the antibody detection generated by the other genotypes.

Serological testing for anti-HDV antibodies has become available in a short period of time due to the advantages of quick, cost-effective and simple operation, but its diagnostic efficacy remains unclear. Therefore, this study aimed to provide a brief meta-analysis of research on the diagnostic efficacy of anti-HDV antibody testing, the results of which could assist in the diagnosis of HDV.

## 2. Materials and Methods

The methods and results of this review are presented according to the Preferred Reporting Items for Systematic reviews and Meta-Analyses statement (PRISMA) [15]. The review protocol was registered on PROSPERO (International Prospective Register of Systematic Reviews) under CRD (Centre of Reviews and Dissemination) report number CRD42022315456.

### 2.1. Search Strategy

The PubMed, Web of Science and Cochrane Library databases were searched from 1 January 1989 to 31 May 2023. The databases were searched by combining the following keywords with the corresponding Medical Subject Headings (MESH) terms: “Hepatitis D”, “hepatitis delta virus”, “serological tests”, “diagnosis”, “Enzyme-Linked Immunosorbent Assays”, “Immunoglobulin G”, “Immunoglobulin M”, “Sensitivity and Specificity” and “efficacy”. Chemiluminescent immunoassay is one of the serological detection methods and has been included in the literature search. However, since chemiluminescent immunoassay is rarely used in detecting serological markers of hepatitis D, it is not listed separately in the literature search. The search terms are shown in Appendix A.

### 2.2. Inclusion and Exclusion Criteria 

We checked the reference lists of the included studies to avoid literature omission. We defined the eligibility criteria as follows: (a) numbers of true positives (TP), false positives (FP), true negatives (TN) and false negatives (FN) available; (b) PCR tests for HDV nucleic acids and HDV antibody tests performed. Studies based on diagnosis by serological tests providing detailed measurements of sensitivity, specificity, summary receiver operating characteristic (SROC) curves, likelihood ratios and areas under the SROC curve (AUCs) were included. However, studies were excluded if (a) the number of cases were <10; (b) the studies were case reports, review articles or meta-analysis articles; (c) repeated studies; or (d) the gold standard was not unclear or not used.

### 2.3. Assessment of Methodological Quality

The quality of each study was assessed using the Quality Assessment of Diagnostic Accuracy Studies 2 (QUADAS-2) tool [16]. Patient selection, index test, reference standard and flow and timing are the four core domains of QUADAS-2. Risk of bias was classified as low, high or unclear for each domain. Two reviewers independently performed the assessment to judge the quality of each study. Disagreement among the reviewers was resolved by discussion.

### 2.4. Data Extraction

The two reviewers who performed the literature search also independently extracted data from the enrolled studies using a predefined data extraction form. Variables extracted from the selected studies included author, year of the study, type of anti-HDV (IgG, IgM or total) and method of antibody detection. The diagnostic characteristics of anti-HDV antibody tests, such as TP, FP, TN and FN, were also extracted.

### 2.5. Data Synthesis

STATA SE with MIDAS commands were used for the meta-analysis. A random effects model was applied to calculate the pooled sensitivity, specificity and diagnostic odds ratio (DOR) with a 95% confidence interval (CI). The presence or absence of heterogeneity was identified through SROC curves. A *p* value < 0.05 was employed to demonstrate a statistically significant association in all analyses.

## 3. Results

### 3.1. Search Results

In total, 1376 articles were identified from the PubMed, Web of Science, Cochrane Library, CNKI (China) and Wan fang (China) databases and other sources; 226 duplicates and 969 articles related to other topics were removed, with 181 articles suitable for abstract screening remaining. A total of 101 articles were excluded following abstract screening. Thus, 80 articles were eligible for full-text screening. Two studies evaluated the performance of three ELISA kits and a new automated assay, LIAISON^®^ XL Murex, for detecting HDV antibodies. The results showed that the evaluated methods had good diagnostic efficacy. However, the evaluation criteria used in these two studies were serological diagnostic results and did not conduct HDV RNA detection; as such, they did not meet the literature inclusion criteria of this research, so they were excluded [17,18]. Ultimately, 12 studies meeting the predetermined inclusion and exclusion criteria were included in this meta-analysis [19,20,21,22,23,24,25,26,27,28,29,30]. Figure 1 shows the PRISMA flow chart of the literature search and selection of studies.

### 3.2. Characteristics of the Included Articles

Table 1 provides the characteristics of the included studies. A total of 12 studies were reviewed, and all were included in the meta-analysis. Three of the included studies were conducted in America, three in Spain and one each in Brazil, China, France, I.R. Iran and Japan. The sample size ranged from 16 to 194, with a total sample size of 1303. Anti-HDV IgM antibody testing was performed in 7/12 studies, anti-HDV IgG antibody testing was performed in 4/12 studies and anti-HDV total antibody testing was performed in 8/12 studies. All the included studies were in the English language, from 1989 to 2023.

### 3.3. Quality Assessments

Figure 2A displays each of the 12 individual QUADAS-2 evaluations, and Figure 2B summarizes the QUADAS-2 assessment. Bias in each study was assessed as “low risk of bias”, “high risk of bias” or “unclear risk of bias”. All studies included in this systematic review had a low risk of bias in the domains of the index test. Because all studies reviewed used HDV diagnosis as the primary objective, no issues relating to applicability of the index test were identified. For the patient selection domain, 50% (6/12) of the assessments concluded a high or unclear risk of bias because these studies were mostly related to a case–control design and did not use consecutive or random sampling. From an applicability perspective, 67% (8/12) were considered low risk for applicability of patient selection, and the remaining studies were assessed as unclear or high risk. For the reference standard domain, we judged the risk of bias as unclear in 75% (9/12) of assessments owing to inadequate details about specimens. In terms of applicability, 83% (10/12) of the studies were classified as low risk in the reference standard, and the remaining 17% (2/12) of studies were classified as unclear risk.

### 3.4. Diagnostic Performance

The results of the random effects meta-analysis are shown in Figure 3. Figure 3D is a summary of the results of Figure 3A–C, intended to evaluate the sensitivity and specificity of serologic detection of HDV antibodies in general. As some of the included literature contain multiple antibody detection data, there are duplicate studies in Figure 3D, but no duplicate data. The pooled sensitivity and specificity for serological tests were 0.99 (95% CI: 0.96–1.00) and 0.90 (95% CI: 0.79–0.96). The sensitivity and specificity were 1.00 (95% CI: 0.00–1.00) and 0.71 (95% CI: 0.56–0.83) for anti-HDV total antibodies, 0.96 (95% CI: 0.83–0.99) and 0.98 (95% CI: 0.82–1.00) for anti-HDV IgM and 0.95 (95% CI: 0.86–0.98) and 0.96 (95% CI: 0.67–1.00) for anti-HDV IgG, respectively. 

SROC curves were generated to indicate overall diagnostic accuracy. The AUC was 0.99 (95% CI: 0.98–1.00) for anti-HDV serological tests, 1.00 (95% CI: 0.99–1.00) for anti-HDV total antibodies, 0.99 (95% CI: 0.98–1.00) for anti-HDV IgM and 0.96 (95% CI: 0.94–0.98) for anti-HDV IgG (Figure 4). Table 2 reports details of the 12 included studies with regard to the sensitivity and specificity of serological tests that measured total, IgM and IgG antibodies. The pooled diagnostic odds ratio, positive likelihood ratio and negative likelihood ratio are shown in Table 3.

## 4. Discussion

HDV is a defective virus that relies on HBV to replicate and spread, so drugs that focus on achieving HBsAg loss and HBV DNA negative at 24 weeks after the end of treatment (HBV functional cure) can achieve both goals—HDV and HBV cure. A treatment that leads to a functional cure for HBV will also lead to the clearance of HDV, so the strategy for HDV treatment can be twofold, directly targeting HDV or, preferably, a functional cure for HBV [31]. Nucleoside analogues (NAs) are a class of antiviral drugs that inhibit the activity of HBV polymerase. Given that the life cycle of HDV is dependent on continued HBV infection and the production of hepatitis B surface antigen, treatment with a functional cure of HBV can lead to HDV eradication. Although NA is an effective therapy for inhibiting HBV replication in patients with CHB, it does not usually lead to loss of HBsAg [32]. A meta-analysis of 29 studies involving 1896 patients reported an incidence of HBsAg disappearance (12 months after NA therapy) of 0.58% per year. Therefore, NA therapy is not effective for HDV clearance. In NA-treated CHB patients, HDV viremia continues to play a key role in HCC development [33]. 

Chronic HDV infection is the most severe form of viral hepatitis in humans and may accelerate liver fibrosis [34,35,36]. Moreover, enduringly detectable HDV viremia has been suggested to lead to a higher rate of progression to liver cirrhosis and hepatic decompensation [35,37]. Japanese researchers have published two studies on HDV antibody screening in HBsAg seropositivity patients, and the results show that compared with seropositivity HDV patients, serum HDV antibody-positive patients have a significantly higher prevalence of cirrhosis, lower prothrombin time, higher prevalence of HIV co-infection and faster progression of liver fibrosis. This result highlights the importance of routine HDV testing [38,39]. An estimated 12 million people worldwide have had HDV infection [40], with a higher prevalence in certain geographic areas and populations. As HDV infection depends on the presence of HBV, EASL, AASLD and APASL guidelines recommend HDV testing for all HBsAg-positive patients chronically infected with HBV [41,42,43]. However, these guidelines are not strictly followed in clinical practice [44], with insufficient access to testing equipment being one of the reasons [45]. The detection methods of HDV mainly include nucleic acid testing and serological tests. A study in France analyzed the results of serological and molecular biological tests for HDV in different laboratories. The results showed that the serological tests were highly consistent among laboratories, while the molecular biological results were not ideal. The researchers pointed out that efforts should continue to improve and standardize quantitative analysis, including the use of the newly launched WHO HDV RNA standards, to promote collaborative clinical research and ultimately optimize patient management [46]. HDV RNA is the current “gold standard” for diagnosing HDV infection, but it has several limitations: (a) PCR testing only detects active HDV RNA replication, which can be suppressed during therapy, (b) the GC content in HDV RNA is as high as 60%, and about 74% of the bases are complementary pairs within the molecule, so the high proportion and complementarity of GC in HDV RNA has brought great technical challenges to HDV amplification and (c) for the wide genetic variability of HDV RNA, there is no fully standardized PCR detection technique until now, and the results from different laboratories are not comparable. As a result, more convenient and cost-efficient serological tests for anti-HDV are used for screening purposes [47]. Nonetheless, the diagnostic efficacy of the serum antibody test reported in earlier studies confused researchers. Therefore, it is of great public health significance to evaluate the diagnostic efficacy of serological tests.

We conducted this systematic review and meta-analysis to evaluate the accuracy of HDV serological diagnosis. The serological methods included in the study included RIA, EIA, ELISA and automatic immunoassay analysis. The pooled sensitivity and specificity of serological tests were 0.99 (95% CI: 0.96–1.00) and 0.90 (95% CI: 0.79–0.96). The pooled sensitivity and specificity of IgM were 0.96 (95% CI: 0.83–0.99) and 0.98 (95% CI: 0.82–1.00), respectively, and those of IgG were 0.95 (95% CI: 0.86–0.98) and 0.96 (95% CI: 0.67–1.00), respectively. These meta-analysis results indicate promising accuracy for IgM detection in diagnosing HDV. The pooled diagnostic performance of IgG was slightly lower than that of IgM. Because we retrieved only four studies on HDV IgG detection, there was no significant difference in the sensitivities and specificities of IgM and IgG among them. We combined the results of IgM and IgG detection, and the resulting pooled sensitivity and specificity were 0.97 (95% CI: 0.89–0.99) and 0.98 (95% CI: 0.88–1.00) (Appendix A), respectively. For serological detection of anti-HDV total antibodies, the pooled sensitivity and specificity were 1.00 (95% CI: 0.00–1.00) and 0.71 (95% CI: 0.56–0.83), respectively. We found that specificities were consistently lower for the total antibody compared with IgM and IgG. Taking the principles of accuracy and simplicity into consideration, we conclude that detection of IgM/IgG is a better choice for HDV diagnosis.

Despite the availability of a large number of highly sensitive and specific hepatitis D antibody test kits, research on the epidemiology of hepatitis D remains challenging on a global scale. First, there is a lack of public awareness, and screening for hepatitis D is not used routinely in clinical practice. Second, standardized methods for HDV testing are still lacking. The quality of kits from different manufacturers varies, and there is little comparability between the preparation and application of standards. In addition, the hepatitis D antibody test has a short window period, and the formation of complexes between antigen and antibody is not easily detected. Therefore, serological test kits with high sensitivity and specificity, common standards and the ability to overcome the short window period of antibodies need to be developed.

## 5. Limitations

This study has a few limitations that must be noted. Firstly, although we have attempted to conduct a comprehensive search of the published studies and include all available trials, we cannot rule out the possibility of missing literature due to the exclusion of non-English articles and conference abstracts. And we did not obtain unpublished data by contacting the authors.

Secondly, this meta-analysis had high heterogeneity, including regarding methodological issues, such as variation in study design, inclusion and exclusion criteria and sampling frameworks. The random effects model was applied to weaken the influence of heterogeneity.

Thirdly, there were limitations in the data available for this analysis. Only four studies involved IgG detection, which could lead to bias. Also, due to the limited data available for each serological test, no subgroup analysis of the different serological tests was performed to derive the source of heterogeneity.

Fourthly, HDV diagnosis is generally based on HDV RNA detection as the gold standard, but this gold standard has some problems. On the one hand, in clinical practice, HDV RNA detection is usually only performed when the HDV serological test is positive, which results in a sensitivity much higher than the actual value. On the other hand, the sensitivity of the dot hybridization method used to detect HDV RNA in early included studies is much lower than that of the RT-PCR method used in recent years. If this standard is used, a higher than actual serological detection sensitivity will be obtained. As there is no fully standardized PCR detection technique to date, results from different laboratories are also not comparable. Since there is still no standard reference, relevant regulatory agencies should develop relevant standards as soon as possible to standardize HDV diagnosis. In addition, the study included a large time span, which included a period of rapid development of serological detection methods. The early literature included relatively ancient serological detection methods, which cannot represent the accuracy of serological diagnosis at the present stage, and therefore led to less reliable results. A large number of recent data are still needed to improve this result and support this conclusion.

## 6. Conclusions

Notwithstanding the above limitations, our results confirm that serological tests have good accuracy in detecting HDV antibodies, especially in anti-HDV IgM and IgG. However, the specific effectiveness of serological diagnosis needs more data support. Overall, serological testing is a good choice for HDV screening. The development of new serological diagnostic tools with higher accuracy and reliability in the diagnosis of HDV is of great importance for laboratory diagnosis, analysis of treatment efficacy, epidemiology and disease control.

## Figures and Tables

**Figure 1 viruses-15-02345-f001:**
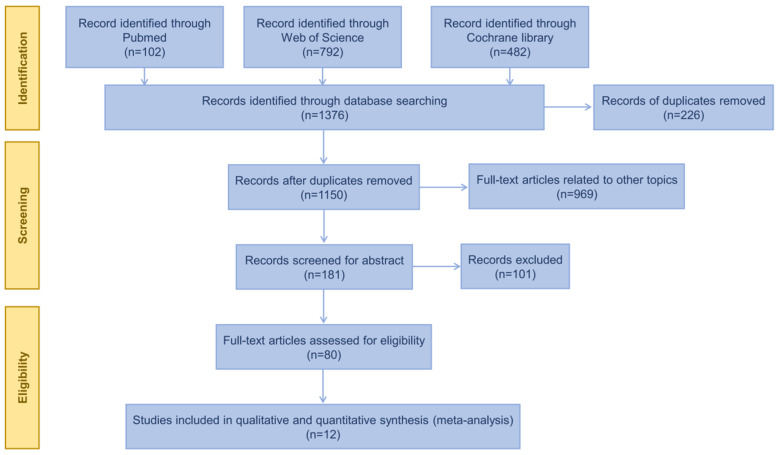
The steps of the literature search and selection.

**Figure 2 viruses-15-02345-f002:**
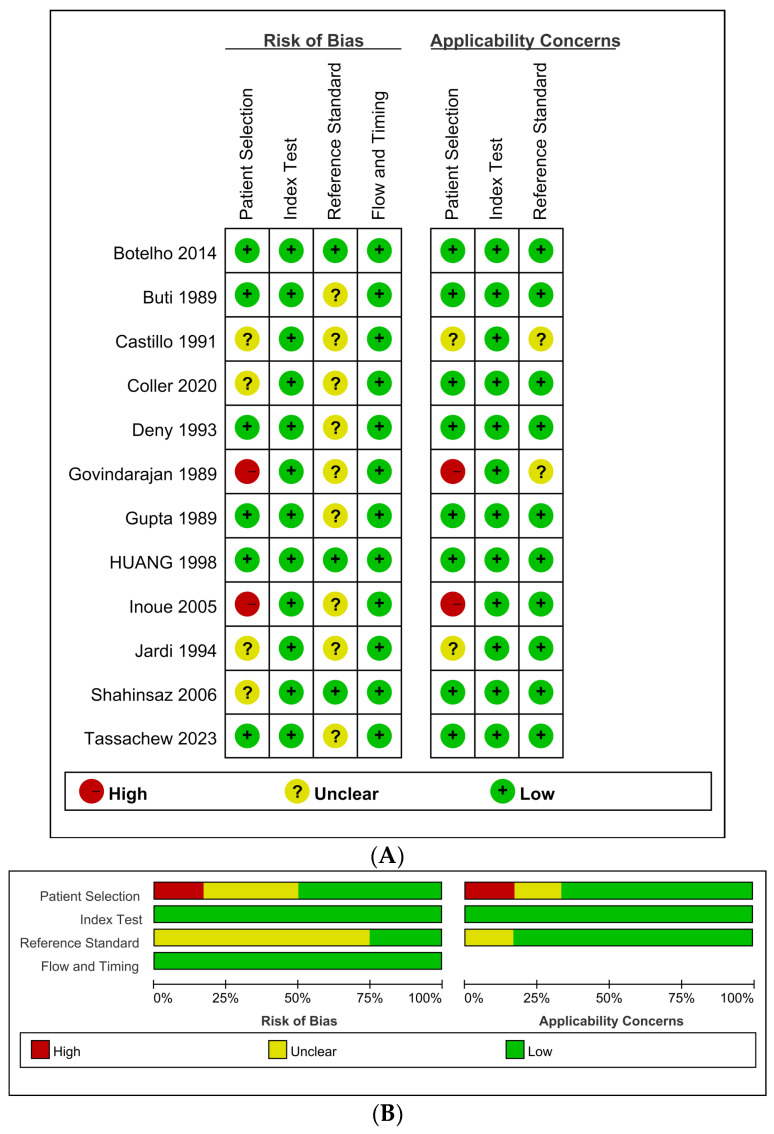
Quality assessment of the study. (**A**) Risk of bias and applicability concerns graph [19,20,21,22,23,24,25,26,27,28,29,30]. (**B**) Risk of bias and applicability concerns summary.

**Figure 3 viruses-15-02345-f003:**
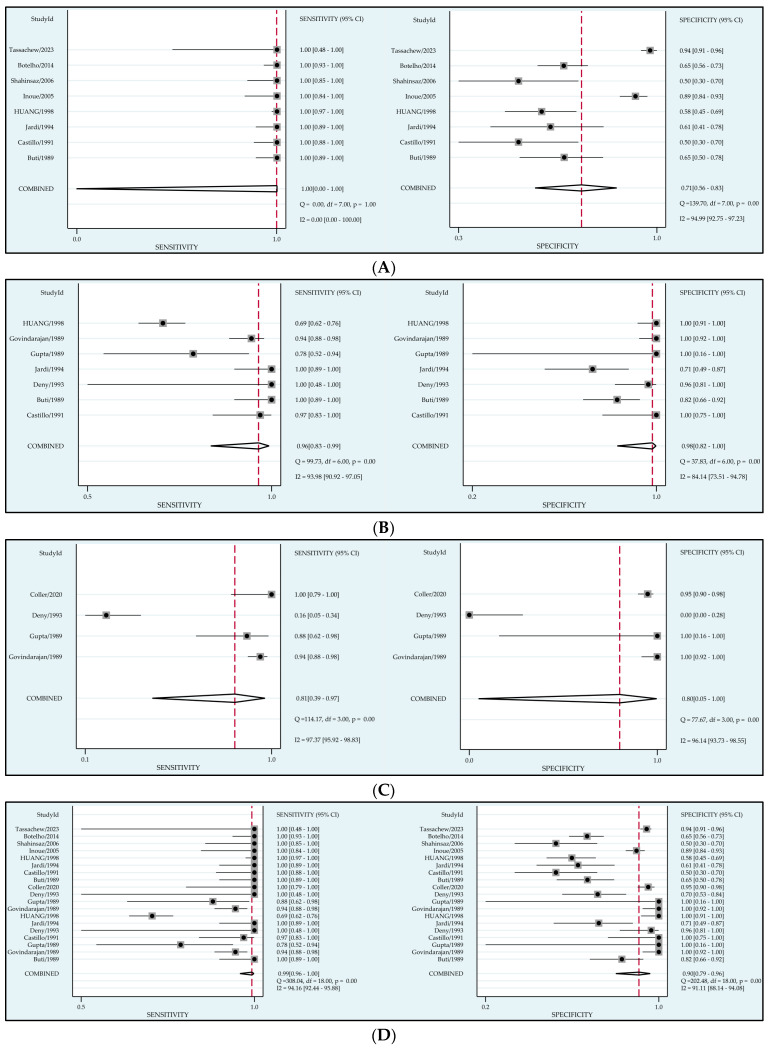
Forest plots of the pooled sensitivity and specificity for anti-HDV antibodies in diagnosis of HDV. (**A**) Total antibody [19,22,24,25,26,27,28,30], (**B**) IgM [19,20,21,22,23,24,25], (**C**) IgG [20,21,23,29], (**D**) SUM [19,20,21,22,23,24,25,26,27,28,29,30].

**Figure 4 viruses-15-02345-f004:**
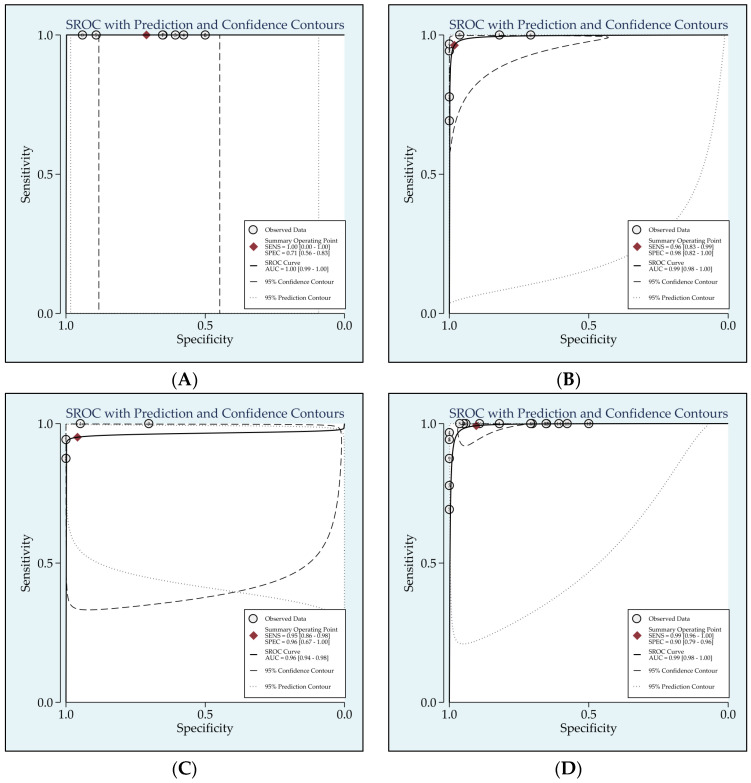
The SROC curves of the serological testing of anti-HDV antibodies. (**A**) Total, (**B**) IgM, (**C**) IgM, (**D**) SUM.

**Table 1 viruses-15-02345-t001:** The main features of the included studies for anti-HDV antibodies in the diagnosis of HDV.

NO.	Author	Country	Year	Number (Cases/Sample Size)	Methods of Detecting HDV RNA	Types of Antibody	Methods of Detecting Antibody
1	Maria Buti [19]	Spain	1989	33/65	Spot Hybridization Test	Total/IgM	RIA/EIA
2	Sugantha Govindarajan [20]	America	1989	99/144	Spot Hybridization Test	IgM/IgG	RIA
3	Sanjeev Gupta [21]	America	1989	14/16	Spot Hybridization Test	IgM/IgG	RIA
4	I. Castillo [22]	Spain	1991	30/43		Total/IgM	RIA/EIA
5	Paul Deny [23]	France	1993	5/31	PCR	IgM/IgG	ELISA/EIA
6	Rosendo Jardi [24]	Spain	1994	33/50	Spot Hybridization Test	Total/IgM	RIA/EIA
7	Yi-Hsiang Huang [25]	China	1998	76/116	RT-PCR	Total/IgM	EIA
8	Jun Inoue [26]	Japan	2005	21/194	RT-PCR	Total	ELISA
9	Leila Shahinsaz [27]	I.R. Iran	2006	23/36	Nested-PCR	Total	ELISA
10	Luan Felipo Botelho-Souza [28]	Brazil	2014	54/140	RT-qPCR	Total	ELISA
11	Kelly E. Coller [29]	America	2020	16/145	RT-qPCR	IgG	Abbott ARCHITECT
12	Yayehyirad Tassachew [30]	Ethiopia	2023	5/323	RT-PCR	Total	ELISA

Note: Only the first author of each study is given. Abbreviations: Total, total antibody; IgG, immunoglobulin G; IgM, immunoglobulin M; PCR, polymerase chain reaction; RT-PCR, reverse transcription-polymerase chain reaction; Nested-PCR, nested-polymerase chain reaction; RT-qPCR, real-time quantitative polymerase chain reaction; RIA, radioimmunoassay; EIA, enzyme immunoassay; ELISA, enzyme-linked immunosorbent assay.

**Table 2 viruses-15-02345-t002:** Individual and pooled sensitivity and specificity by immunoglobulin class detected.

Method and Studies	TP	FN	Sensitivity (95% CI^a^)	TN	FP	Specificity (95% CI)
Total (*n* = 8)						
Maria Buti	33	0	1.00 (0.89–1.00)	32	17	0.65 (0.50–0.78)
I. Castillo	30	0	1.00 (0.88–1.00)	13	13	0.50 (0.30–0.70)
Rosendo Jardi	33	0	1.00 (0.89–1.00)	17	11	0.61 (0.41–0.78)
Yi-Hsiang Huang	137	0	1.00 (0.97–1.00)	41	30	0.58 (0.45–0.69)
Jun Inoue	21	0	1.00 (0.84–1.00)	173	21	0.89 (0.84–0.93)
Leila Shahinsaz	23	0	1.00 (0.85–1.00)	13	13	0.50 (0.30–0.70)
Luan Felipo Botelho-Souza	54	0	1.00 (0.93–1.00)	86	46	0.65 (0.56–0.73)
Yayehyirad Tassachew	5	0	1.00 (0.48–1.00)	318	20	0.94 (0.91–0.96)
Pooled	336	0	1.00 (0.00–1.00)	693	171	0.71 (0.56–0.83)
IgM (*n* = 7)						
Maria Buti	33	0	1.00 (0.89–1.00)	32	7	0.82 (0.66–0.92)
Sugantha Govindarajan	99	6	0.94 (0.88–0.98)	45	0	1.00 (0.92–1.00)
Sanjeev Gupta	14	4	0.78 (0.52–0.94)	2	0	1.00 (0.16–1.00)
I. Castillo	30	1	0.97 (0.83–1.00)	13	0	1.00 (0.75–1.00)
Paul Deny	5	0	1.00 (0.48–1.00)	26	1	0.96 (0.81–1.00)
Rosendo Jardi	33	0	1.00 (0.89–1.00)	17	7	0.71 (0.49–0.87)
Yi-Hsiang Huang	137	61	0.69 (0.62–0.76)	41	0	1.00 (0.91–1.00)
Pooled	351	72	0.96 (0.83–0.99)	176	15	0.98 (0.82–1.00)
IgG (*n* = 4)						
Sugantha Govindarajan	99	6	0.94 (0.88–0.98)	45	0	1.00 (0.92–1.00)
Sanjeev Gupta	14	2	0.88 (0.62–0.98)	2	0	1.00 (0.16–1.00)
Paul Deny	5	0	1.00 (0.48–1.00)	26	11	0.70 (0.53–0.84)
Kelly E. Coller	16	0	1.00 (0.79–1.00)	129	7	0.95 (0.90–0.98)
Pooled	134	8	0.95 (0.86–0.98)	202	18	0.96 (0.67–1.00)
SUM (*n* = 19)	821	80	0.99 (0.96–1.00)	1071	204	0.90 (0.79–0.96)

CI^a^, confidence interval.

**Table 3 viruses-15-02345-t003:** Summary table of the diagnostic accuracy of total antibody, IgM, IgG and IgM/IgG for HDV infection.

	Sensitivity (95% CI)	Specificity (95% CI)	DOR^a^ (95% CI)	Lrpos^b^ (95% CI)	Lrneg^c^ (95% CI)	AUC^d^ (95% CI)
Total	1.00 (0.00–1.00)	0.71 (0.56–0.83)	——	3.5 (2.1–5.6)	——	1.00 (0.99–1.00)
IgM	0.96 (0.83–0.99)	0.98 (0.82–1.00)	1399 (157–12,430)	52.8 (4.9–573.1)	0.04 (0.01–0.19)	0.99 (0.98–1.00)
IgG	0.95 (0.86–0.98)	0.96 (0.67–1.00)	451 (27–7451)	23.0 (2.2–241.2)	0.05 (0.02–0.16)	0.96 (0.94–0.98)
IgM/IgG	0.97 (0.89–0.99)	0.98 (0.88–1.00)	1277 (256–6363)	44.2 (7.6–256.0)	0.03 (0.01–0.11)	0.99 (0.98–1.00)
SUM	0.99 (0.96–1.00)	0.90 (0.79–0.96)	1199 (253–5675)	10.3 (4.6–23.1)	0.01 (0.00–0.05)	0.99 (0.98–1.00)

DOR^a^, diagnostic odds ratio; Lrpos^b^, positive likelihood ratio; Lrneg^c^, negative–positive likelihood ratio; AUC^d^, area under the curve.

## Data Availability

Data is contained within the article or Appendix A.

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
