# Peer review of "Diagnostic Efficacy of Serological Antibody Detection Tests for Hepatitis Delta Virus: A Systematic Review and Meta-Analysis"

_viruses, 2023, doi:10.3390/v15122345_

Round 1

Reviewer 1 Report

Comments and Suggestions for Authors

This is a very detailed meta-analysis of the accuracy of existing anti-HDV assays. I have just some suggestions for the Discussion:

1) Can the Authors discuss the fact that as of today there is no standard reference assay, and that this should be a priority for regulatory agencies?

2) Do HDV genotypes influence the accuracy of anti-HDV testing? 

3) Why the CLIA were not included in the analysis?

4) Can the Authors explain why all reference studies are mentioned twice in Figure 3D?

4) can the Author mention and comment upon the studies by Brichler S, et al. J Clin Microbiol 2014;52:1694-1697; Lin GY, et al. Virol J 2020;17:76.; and Rocco C, et al. Diagn Microbiol Infect Dis 2019;95:114873

Comments on the Quality of English Language

Minor language editing: please be consistent, names which are abbreviated should be mentioned in extenso again (Hepatitis D Virus etc.)

The reference list requires major attention: several names of Authors are misspelled

Reviewer 2 Report

Comments and Suggestions for Authors

1.    Authors should discuss whether agents leading to functional cure of HBV are ideal for both HBV and HDV.

2.    Give the several comments on that HDV viremia played a crucial role in HCC development in CHB patients who underwent NA therapy.

3.    Add the reference: Sasaki T, et al. Hepatol Res. 2023 Oct;53(10):960-967. doi: 10.1111/hepr.13936. PMID: 37332115; Omata M, et al. Hepatogastroenterology. 1985 Oct;32(5):220-3.PMID: 4077011.
